# A Brief Theory of Epidemic Kinetics

**DOI:** 10.3390/biology9060134

**Published:** 2020-06-22

**Authors:** François Louchet

**Affiliations:** Grenoble Institute of Technology and Laboratoire de Glaciologie et de Geophysique de l’Environnement, CNRS, Grenoble University (Retired), 38410 St Martin d’Uriage, France; francoislouchet38@gmail.com

**Keywords:** epidemic, COVID-19, contamination kinetics, herd immunity, dynamical systems, reproductive rate, critical state, attractor, stable cycle, chaos

## Abstract

In the context of the COVID-19 epidemic, and on the basis of the Theory of Dynamical Systems, we propose a simple theoretical approach for the expansion of contagious diseases, with a particular focus on viral respiratory tracts. The infection develops through contacts between contagious and exposed people, with a rate proportional to the number of contagious and of non-immune individuals, to contact duration and turnover, inversely proportional to the efficiency of protection measures, and balanced by the average individual recovery response. The obvious initial exponential increase is readily hindered by the growing recovery rate, and also by the size reduction of the exposed population. The system converges towards a stable attractor whose value is expressed in terms of the “reproductive rate” *R*_0_, depending on contamination and recovery factors. Various properties of the attractor are examined, and particularly its relations with *R*_0_. Decreasing this ratio below a critical value leads to a tipping threshold beyond which the epidemic is over. By contrast, significant values of the above ratio may bring the system through a bifurcating hierarchy of stable cycles up to a chaotic behaviour.

## 1. Introduction

In the beginning of 2020, we started experiencing the so-called COVID-19, an unprecedented epidemic, at least since the “Spanish Flu” in 1918–1919, and more recently (and to a lesser extent) SARS-1 in 2003. Despite considerable scientific and medical advances since then, most countries seem to have been overtaken by the epidemic development, and unable to predict, even qualitatively, when and in which way it may recede and possibly disappear. Understanding the kinetic evolution of infectious diseases is a long-lasting challenge. As analysed in [1], epidemic modelling may involve either statistical or so-called “mechanistic” approaches. The former ones, through extrapolation of curves previously fitted on available data, aim at providing short-term forecasts, limited by data uncertainties, whose effects are enhanced by the highly non-linear nature of involved mechanisms. The latter are of quite a different nature. They aim at exploring fundamental bases and long-term trends of epidemic evolution, thanks to their propensity for taking into account non-linear effects, and help explore the possible role of so-called “control parameters”.

As early as at the beginning of the 20th century [2,3], a basis for theoretical modelling was established, using such “mechanistic” models, particularly in terms of population groups experiencing the different infectious stages of the epidemic. Such approaches were subsequently extended and refined, incorporating, for instance, host demography and migrations, as well as diffusion and mutation of infectious agents [4]. A quite recent example [5] is a detailed analysis of the COVID-19 outbreak in Wuhan. A set of differential equations describes the evolution of the four traditional types of populations, Susceptible–Exposed–Infected–Recovered (SEIR). The equations are parametrized using reported cases, and solved numerically in order to estimate the number of unreported cases and to predict the epidemic evolution.

On the other hand, it is well known that a fundamental characteristic of the evolution of complex non-linear systems is their extreme sensitivity to initial conditions, as first discovered by Henri Poincaré [6]. In the case of the evolution of complex systems, and more particularly of epidemic dynamics, this sensitivity makes difficult both conventional long-term predictions and standard system stability analysis [7]. However, due to the highly non-linear nature of the dynamics, simple models based on the Theory of Dynamical Systems, which are not necessarily quantitative and don’t necessarily need to be fitted on experimental data at this stage, may reveal the possible occurrence of specific behaviours, which standard local stability analyses or digital resolution of differential equations may not be able, in essence, to detect as such. This is the case, for instance, for attractors, bifurcations and deterministic chaos. Such models are also useful to better understand the fundamental role of basic control parameters of epidemic expansion.

The present study belongs to this last category. In this spirit and on the basis of the Theory of Dynamical Systems, we propose here a simple and general theoretical model for expansion and possible recession of contagious epidemics. Using assumptions fairly similar to SEIR ones, we shall look for the possible attractor(s) of the system, the influence of its finite size on their position, and the necessary conditions to bring the basic reproductive factor *R*_0_ below 1. The difference between the basic and effective reproductive factors will also be discussed in terms of the attractor properties. Finally, the possible occurrence of multi-stable cycles and chaotic behaviour during a sudden change in control parameter values will be investigated.

Calculations were made from analytical equations mentioned in the text, and figures produced, using basic Excel software.

## 2. Contamination Kinetics

We study an epidemic starting at a given time and place, from a contamination of a human being, following, for instance, a mutation of a virus of animal origin. The infection is gradually transmitted to other people around, resulting in a contaminated area (cluster).

By contrast for instance with condensation of water vapour into droplets, the “energy” of the interface that separates the cluster from the remainder of the population is zero. There is no physical restriction against cluster growth, which is only governed by the exchange kinetics of infectious agents. The critical size in the sense of nucleation-growth processes is therefore also zero, which means that a cluster made of only two individuals is already unstable and may grow spontaneously. The interfacial energy being zero, the cluster is also likely to expand in space with a fractal structure (e.g., romanesco broccoli structure), which favours exchange mechanisms between contagious and exposed people through an optimization of the exchange interface.

Let us consider a total population of N individuals, a number N_C_ of them being contagious at a given time *t*. Neglecting the (nevertheless unfortunate) number of fatalities (about 2% in the case of COVID-19), the remainder (*N*-*Nc*) is made of exposed people (i.e., liable to infection), but also of individuals protected by long-term immunity, resulting from previous contamination and recovery (still debated for COVID-19), or from vaccination (if available and efficient).

The effective number of exposed people may therefore be written $\left(1-\xi \right)\left(N-N_{C}\right)$, where *ξ* is the proportion of immunized people. We assume that the contamination rate (*dN_C_*/*dt*)^+^ is proportional to the product $N_{C}\left(1-\xi \right)\left(N-Nc\right)$ of the number of contagious individuals by that of exposed ones in close contact with them at a given time. It is also taken proportional to the duration *δt* of the contact, to the turnover rate *ν* of exposed/contagious pairs, and inversely proportional to a protection efficiency factor *p* (e.g., mask wearing, lockdown measures, and so forth).

The contamination rate is also controlled by the activity of the infectious agent. In the case of viral respiratory tract diseases, it is sometimes stated that viruses develop in winter season, and get asleep, or at least less active, in summer. This might be due to enhanced UV radiation and air dryness in summer, but this point is still debated in the case of COVID-19. In this state of uncertainty, we shall introduce this supposed influence through a seasonal virus activity factor *S* taken equal to 1 for “season-insensitive” epidemics, and otherwise proportional to a “season parameter” *S* varying sinusoïdally with time:(1)$$S=\frac{1}{2}\left[1+\cos\left[2\pi \left(t/12\right)\right]\right]\;=\;\frac{1}{2}\left[1+\cos\left[\pi t/6\right]\right]$$
where *t* is the time in months, starting in December in the northern hemisphere. Thus, the season parameter *S* would decrease from 1 in December down to 0 in June, and start increasing again in autumn, or alternatively kept equal to 1 if virus activity is constant throughout the year.

The contamination evolution equation can be written:(2)$$\begin{array}{l} {\left(\frac{dN_{c}}{dt}\right)}^{+}=\left(\nu \;\delta \;t/p\right)\;S\;N_{C}\left(1-\xi \right)\left(N-Nc\right)\\ =\;C\;N_{C}\left(1-\xi \right)\left(N-Nc\right) \end{array}$$
where *C* = (*ν δt*/*p*)*S* is a global “contamination factor”.

## 3. Recovery Kinetics and Global Evolution

We have now to introduce a recovery term. The recovery rate is taken to be proportional to the number *N_C_* of contagious individuals, and is controlled by a recovery parameter *D*, which may be understood as the average reciprocal duration (1/*τ*) of the disease hosted by individuals.
(3)$${\left(\frac{dN_{c}}{dt}\right)}^{-}\;=\;DN_{c}$$

The global evolution equation is thus obtained by combining Equations (2) and (3), as shown in Figure 1:(4)$$\begin{array}{l} \frac{dN_{c}}{dt}\;=\;{\left(\frac{dN_{c}}{dt}\right)}^{+}\;-\;{\left(\frac{dN_{c}}{dt}\right)}^{-}\\ =\;C\;N_{c}\left(1-\xi \right)\;\left(N\;-\;N_{c}\right)\;-\;DN_{c}=\left[C\;\left(1-\xi \right)\;N-D\right]\;N_{c}-C\left(1-\xi \right)\;{N_{c}}^{2}\\ =\alpha \;N_{C}-\beta \;{N_{C}}^{2} \end{array}$$
with *α* = [*C*(1 − *ξ*)*N* − *D*] and *β* = *C*(1 − *ξ*).

Figure 1 shows the parabolic variations of *d*(*N_C_*/*N*)/*dt* as a function of *Nc*/*N* for three different cases. For high contamination, slow recovery and low immunity (overcritical case), the parabola intersects the horizontal axis at two *Nc*/*N* values, *Nc*/*N* = 0 and *Nc*/*N* = *N_C_**/*N*. called “fixed points”. The derivative *d*(*Nc*/*N*)/*dt* being zero at such points, *Nc*/*N* does not vary with time. However, slight variations of *Nc*/*N* around *N_C_**/*N* or around zero would result in quite different behaviours. Starting from A, a fluctuation of *Nc*/*N* to the left would bring the system in a region where *d*(*Nc*/*N*)/*dt* is positive, which would push *Nc*/*N* back to the fixed point “A”. Conversely, a fluctuation towards the right, where the derivative is negative, would bring the system to the left, back to “A” again. “A” is called an attractor. For opposite reasons, any fluctuation of the system starting from O (and necessarily to the right) would drive it to the positive derivative zone, up to “A” again. “O” is a repulsor.

The *N_C_* value at the attractor A is easily found from Equation (4), solving the equation:(5)$$\frac{dN_{c}}{dt}\;=\;\left[C\left(1-\xi \right)N-D\right]N_{c}-C\left(1-\xi \right)\;{N_{c}}^{2}=0$$
whose non-zero solution is:$$N_{C}*\;=\;\frac{C\left(1-\xi \right)N-D}{C\left(1-\xi \right)}=\frac{\alpha }{\beta }$$
or equivalently
(6)$$\;\frac{N_{C}*}{N}\;=\;1-\frac{D}{CN\left(1-\xi \right)}=1-\frac{1}{R_{0}}$$
where we define *R*_0_ as *R*_0_ = *CN*(1 − *ξ*)/*D*, which depends on the long-term immunity factor *ξ*, on contamination and recovery factors, and also on the total population size *N*.

At very low *Nc* values (early stages of epidemic) the quadratic term of Equation (4) is negligible as compared to the linear term. Equation (4) may be approximated by:(7)$$\frac{dN_{c}}{dt}\;=\;\left[CN\left(1-\xi \right)-D\right]\;N_{C}$$

Integration of this equation shows that in this case *N_C_* increases exponentially with time, a least as long as [*CN*(1 − *ξ*) − *D*] is positive (i.e., *R*_0_ > 1). This is actually observed in early stages of COVID-19 in most countries, as shown in [8], and represented by straight lines if the number of infected people is plotted vs. time in a semi-logarithmic scale.

Coming back to the general case, Equation (6) means that the proportion of infected people in the current state (attractor) is 1 − 1/*R*_0_, and the proportion of non-infected individuals, including both susceptible and immunized ones, is clearly 1/*R*_0_.

It is obvious from Equation (6) that, in the extreme case where *R*_0_ tends to infinity (infinite contamination factors and (or) no recovery), the steady state proportion *Nc**/*N* of contaminated people tends to 1 (100% infected). On the contrary, the fraction *Nc**/*N* of contaminated people at the attractor A is reduced by a decrease in the contamination factor *C* and an increase in the recovery rate *D* and in the long-term immunity *ξ*, as intuitively expected, and eventually reaches zero when *R*_0_ = 1.

At the attractor, despite the fact that the “basic reproductive rate” *R*_0_ = *CN*(1 − *ξ*)/*D* is larger than 1, there is a balance between newly contaminated and recovering people, which may appear at first sight to be equivalent to a 1 to 1 transmission of the epidemic, suggesting a reproductive rate equal to 1. Even so, the epidemic is not over at this stage. The literature appears as somewhat unclear on this point, probably related to a wide variety of definitions of the reproductive rate [9,10,11]. Actually, the situation at the attractor A does not correspond to *R*_0_ = 1, but instead to *R** = 1, *R** being the “current (or effective) reproductive number”, defined as the ratio of the instantaneous number of new infections by that of fresh recoveries. It is often stated in the literature that in this situation, a single individual contaminates a single other one on average, due to a reduction of the number of exposed people. This is often referred to as the “herd (or global or collective) immunity” threshold [9]. Such a terminology may suggest that contamination of a fraction 1 − 1/*R*_0_ of the population (which is actually the case at the attractor A) should provide a total immunity to the whole population.

The situation is actually slightly more subtle. Let us start with a given value of *R*_0_ larger than one, and observe the epidemic evolution from its very beginning (*N_C_* = 0). Equation (4) shows that the epidemic growth rate is a combination of linear and quadratic terms in *N_C_*. Contamination and recovery kinetics behave differently. The initial growth rate (Equation (7)) is linear in *N_C_*, giving the exponential growth discussed above, but due to such competing kinetics, the “growth rate” readily slows down, goes through a maximum, and eventually converges to zero at the attractor. During this whole process, the contamination rate remains larger than the recovery one, but their difference vanishes as the attractor is approached. The attractor is actually a current, stable but dynamical steady state at which the number of new infections and the number of recoveries in a population balance each other, a fraction of the recovered people becoming immune [12]. Recovering people are continuously replaced by newly contaminated ones, a fraction of them being always vulnerable to death. The epidemic is definitely not over and would not die out spontaneously in this stable state, in contrast with other interpretations of the effective reproductive number [13,14]. The classical representation of the effective reproductive rate in which a single individual directly contaminates a single exposed one surrounded by several immunized other ones seems inadequate. At this stage, the apparent 1 to 1 transmission is actually a balance between contamination and recovery kinetics. This is a key point.

The current proportion of sick people *Nc**/*N* can be reduced if the attractor A is gradually shifted to the left through a reduction of *R*_0_. However, it is worth noting that, during this process, *R** always remains equal to 1 whatever the position of A. This operation would merely decrease the flux of sick people travelling through the “attractor box”, and as a consequence reduce the current occupancy rate of hospital beds (which is already not so bad for lack of anything better).

By contrast, the green parabola in Figure 1 illustrates the limiting case where contamination is low enough and (or) recovery factor strong enough to bring the attractor A to the origin O. Equation (6) shows that the attractor on the green curve has merged with the repulsor O (this is actually a particular type of bifurcation) when *Nc**/*N* = 0, giving:(8)$$R^{*}=R_{0}=\frac{CN\left(1-\xi \right)}{D}=1$$
or equivalently *Nc**/*N* = 1 − 1/*R*_0_ = 0. In this case, *dN_C_*/*dt* becomes negative everywhere except at O, and O is now an attractor, corresponding to a number of contaminated people equal to zero from the very first attempt of the epidemic development.

This situation where *R*_0_ = 1 is equivalent to the critical transition in a nuclear reaction. In this critical state, only one of the two neutrons produced by every fission of a Uranium nucleus is able (in average) to trigger the fission of another Uranium nucleus, and so forth. For *R*_0_ < 1 (undercritical situation), the system remains under control, whereas for *R*_0_ > 1 (overcritical situation), it diverges and the nuclear explosion takes place.

In the epidemic case, for *R*_0_ > 1 (large contamination, low recovery), the number of infected people starts increasing exponentially with time (top blue line in Figure 1), but its growth (“explosion”) is gradually hindered by the increasing recovery rate, and also by the decreasing available amount of exposed individuals, at least if a significant proportion of the population is infected (which does not seem to be the case at this stage for the COVID-19). In a same way, *R*_0_ < 1 corresponds to the undercritical case: every chain reaction gradually slows down and eventually dies out. The epidemic expansion is over as long as *R*_0_ is permanently kept below 1. If not, a new epidemic would be liable to start again somewhere and develop, unless long-term immunity has fully developed, or efficient vaccines have been produced meanwhile.

This behaviour contrasts sharply with exposure to chemical or radioactive pollutants, for which there is no amplification by chain reactions [15]. This last process is linear. Protecting from pollution for instance 20% of a population reduces the contaminated population by 20%. In this case, each individual is responsible for his (her) own health, whereas in the case of contagious epidemics, each individual protection contributes to protection of other people. Epidemic contamination is a chain reaction. The gradual decrease of *Nc**/*N* as *R*_0_ goes down, followed by a threshold at *R*_0_ = 1 after which the epidemic is over, is a typical non-linear effect.

It is worth noting, from Equation (6), that an epidemic propagating in a larger population *N* would require a more stringent limitation of *C*(1 − *ξ*)/*D* to be controlled (in particular stronger protection measures, and vaccination campaigns), as intuitively expected again from the non-linear nature of contamination kinetics. In other words, provisional confinement of people in smaller areas would help in controlling the epidemic, regardless of the obvious associated drawbacks.

## 4. Convergence towards the Attractor, a Possible Route to Chaos?

We shall now investigate the different possible types of convergence to the attractor A shown in Figure 1. The system might indeed converge to the attractor either monotonically, or to a limit cycle, or even possibly experience a transition to a chaotic behaviour [16,17,18].

We start from the global evolution equation (Equation (4)), written for the sake of simplicity:(9)$$dx/dt=\alpha \;x-\beta \;x^{2}$$
with *α* = *CN*(1 − *ξ*) − *D* and *β* = *C*(1 − *ξ*).

Continuous as well as discrete solutions of equations similar to (9), known as “logistic equations”, were proposed a long time ago by Verhulst [19,20] to study population dynamics. Here, in order to investigate the system convergence to the attractor, we shall use the associated “logistic maps”, a simple and quite efficient technique to solve this type of problem.

We discretize Equation (9) into a recursive relation considering finite time steps *δt* = 1 corresponding to steps *δx* for *x*. Starting from an initial value *x*_0_, the evolution of *x* with time can be obtained by the recursive relation:(10)$$\begin{array}{l} x_{n+1}=x_{n}+\delta \;x_{n}\\ =x_{n}+\left(\alpha \;x_{n}-\beta \;{x_{n}}^{2}\right)\\ =\left(1+\alpha \right)\;x_{n}-\beta \;{x_{n}}^{2} \end{array}$$
associated with a “functional relation”:(11)$$f\left(x\right)=\left(1+\alpha \right)\;x-\beta \;x^{2}$$

The fixed points of the system are obtained setting *x_n_*_+1_ = *x_n_*, or equivalently:(12)$$f\left(x\right)=x=\left(1+\alpha \right)\;x-\beta \;x^{2}$$
giving:*x* = 0 and *x* = *x** = *α*/*β*(13)
which is equivalent to Equations (5) and (6).

It is obvious from Equation (12) that fixed points can be represented in a *f*(*x*) vs. *x* graph by the intersections of the *f*(*x*) curve with the diagonal line with unit slope, as shown for instance in Figure 2 and Figure 3.

From Equation (11), the slope *f*′(*x*) of *f*(*x*) is:(14)f′(x)=(1+α)−2β x

Using Equations (13) and (14), we obtain the slope at the fixed points:*f*′(0) = 1 + *α*, and *f*′(*x**) = 1 − *α*(15)

We shall now use the recursive relation (Equation (10)) to determine the various possible behaviours of the system as it approaches the fixed points.

The first example corresponds to *α* = 0 and *β* = 1. This is the critical case (green curve in Figure 1). On the logistic map of Figure 2, *x_n_* values are represented on the horizontal axis, and *x_n_*_+1_ ones on the vertical axis. Starting the iteration from any *x*_0_ value on the horizontal axis (*x*_0_ = 0.6 for instance in Figure 2), we draw a vertical line. It intersects the red parabola giving *x*_1_ = 0.24. A horizontal line drawn from this intersection to the blue diagonal with unit slope transfers *x*_1_ back to the horizontal axis, from which a second iteration is performed, and so forth. The successive *x_n_* values obtained by such iterations are represented by the thin staircase line, that eventually converges to the origin O which is (in this example), the unique fixed point. This is the critical case, corresponding to *R*_0_ = 1.

The second example (Figure 3) corresponds to *α* = 0.9, *β* = 1. In this case, there are two fixed points as shown by Equation (13). Starting from any *x*_0_ value, the iteration converges to the fixed point at *x** = *α*/*β* = 0.9, which is an attractor, whereas O is now clearly a repulsor. In addition, the trajectory to *x** is monotonic as in Figure 2.

The third example (Figure 4) corresponds to *α* = 2.1, *β* = 1. Starting from *x*_0_ = 0.3 for instance (black lines), the system does not converge to *x**, but instead to a so-called “limit cycle”, for which *x* oscillates between two different values on both sides of the attractor (bi-stable state). Starting from *x*_0_ = 2.8 (green lines), the system also converges to the same limit cycle, though after a larger number of iterations. This kind of result is geometrically obvious as soon as the slope of the parabola at *x** becomes less than (−1). In Figure 4, the slope at *x** is indeed (−1.1 < −1). It can be shown [9,10,11] that decreasing further *f*′(*x**) (as *α* goes up) may drive the system through a bifurcating hierarchy of multi-stable cycles that eventually leads to a tipping point and a chaotic behaviour.

Since *α* = *C*(1 − *ξ*)*N* − *D* and *β* = *C*(1 − *ξ*), and in agreement with the three above examples, such a trajectory toward deterministic chaos is obviously favoured by increasing values of $\frac{\alpha }{\beta }=N-\frac{D}{C\left(1-\xi \right)}\;=\;N\left(1-\frac{1}{R_{0}}\right)$, i.e., a high contamination factor *C*, a low recovery rate *D*, a low immunity *ξ*, and also a large total population *N* in the concerned area. A possible consequence of this result is that a sudden decrease of protection factors, as for instance during a rapid release of containment measures that would bring *R*_0_ to a value well above 1, is liable to trigger strong instabilities that might be difficult to control, since needs for emergency hospital beds may suffer large periodic or, even worse, chaotic variations. In this case, a temporary subdivision of infected areas into smaller isolated ones may bring the system back to a more stable and manageable state. In any case, the post-crisis release of protection policies should be conducted in a gradual and controlled manner. Other possible consequences of such a route to chaos are worth being studied.

## 5. Conclusions

The present study is not a statistical model. It does not intend either to produce short or middle-term predictions. It only aims at analysing in detail the specific role of epidemics control parameters. For this purpose, as such processes are highly non-linear, we applied standard methods used in the Theory of Dynamical Systems.

As infection extends through exchanges of infectious agents between contagious and exposed people, the net increase rate *dN_C_*/*dt* of contagious people at a given time results from a competition between two terms: (i) a contamination factor proportional to both the number of contagious individuals and the number of exposed ones, and (ii) a decay factor related to recovery kinetics of contaminated individuals.

The contamination factor involves the duration and turnover frequency of pair contacts, the protection efficiency (e.g., lockdown measures, mask wearing), a long-term immunity factor, and the possible seasonal variations of the virulence of infectious agents. The recovery factor is controlled by the reciprocal average duration of the infected state of individuals. At constant values of control parameters, the system converges to a steady state attractor at which contamination and recovery factors balance each other.

Various properties of the attractor were then thoroughly investigated, more particularly through responses to changes in the values of control parameters.

The smaller the contamination factor and the larger the recovery and the long-term immunity factors are, the lower will be the steady state proportion *Nc**/*N* of contagious people at a given time, as intuitively expected. However, this would definitely not be the end of contamination events. Indeed, *Nc** being the current number of contagious people at time *t*, recovering people are continuously replaced by an equal number of newly contaminated ones in this stage.

This 1 to 1 transmission ratio at the attractor corresponds to a “current” or “effective” reproductive factor *R** equal to 1, in spite of a *R*_0_ value larger than 1. As a consequence, an “imposed” decrease of *R*_0_ through lockdown or vaccination measures for instance would not change the *R** value at the attractor, which will stay equal to 1 regardless of the attractor position, but it would reduce the flux of sick people travelling through the “attractor box”, or in other words the current number of sick people, that would help manage the occupancy of hospital beds. The epidemic would really be under control when the reduction of *R*_0_ would result in a situation where A and O would merge, i.e., for *R*_0_ = *R** = 1.

By contrast, decreasing further the reproductive number *R*_0_ may lead to a tipping threshold beyond which the epidemic is actually over. This would remain true as long as *R*_0_ is kept below 1, i.e. as long as protection measures are enforced, and (or) as long as the long-term immunity of the population (possibly helped by vaccination) has increased enough to be able to take over the lack of protection. 

On the contrary, a sudden change in *R*_0_ value may shift the attractor apart from the actual position of the system. We show that, if the system is driven towards the new attractor position by a too large restoring force, for instance after a too rapid release of confinement or other protection measures, it would be brought through a bifurcating hierarchy of stable cycles to a chaotic behaviour, whose management would resultingly be problematic.

## Figures and Tables

**Figure 1 biology-09-00134-f001:**
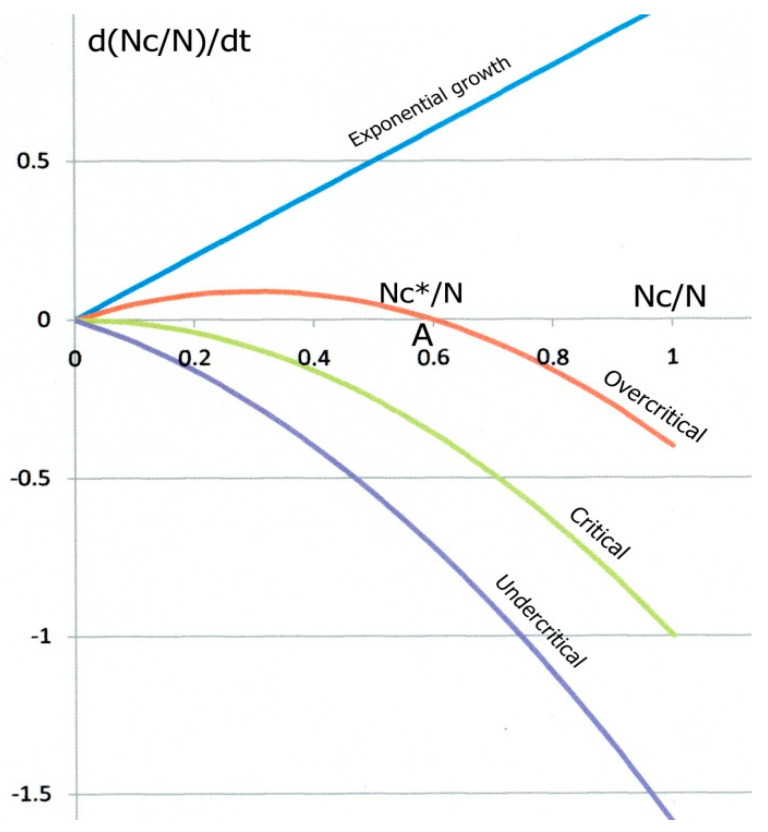
Theoretical epidemic growth kinetics. The variation rate *d*(*Nc*/*N*)/*dt* (Equation (4) normalized by *N*) of the proportion *Nc*/*N* of contagious people vs. *Nc*/*N*. In the overcritical case (red curve), a contagious individual contaminates more than another one on average. The red parabola intersects the *Nc*/*N* axis at a repulsor O and an attractor A. At the attractor, the steady state value of *Nc*/*N* is *Nc**/*N* = 1 − 1/*R*_0_. Decreasing *R*_0_ gradually reduces the *Nc**/*N* value down to zero (critical state, green curve), at which A and O merge for *R*_0_ = 1. Reducing *R*_0_ below 1 brings the system to an undercritical state (blue curve) for which a single contagious individual contaminates less than an exposed one on average; in such conditions, the epidemic eventually dies out.

**Figure 2 biology-09-00134-f002:**
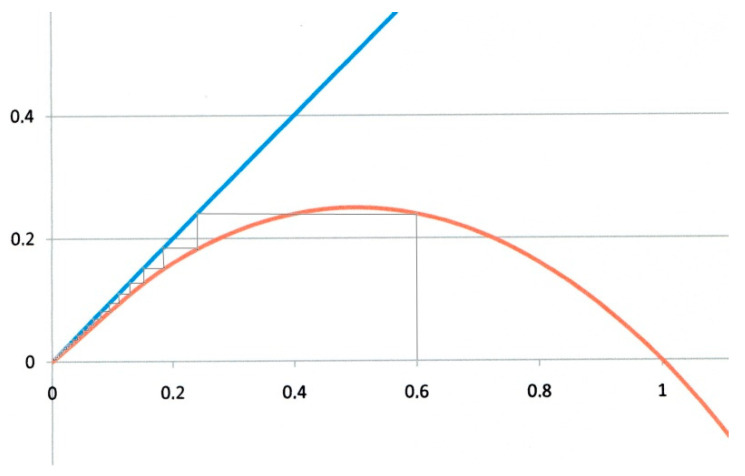
Logistic map for *α*= 0 and *β* = 1. Critical case. Starting from any *x*_0_ value (0.6 in the figure) the system eventually converges to the origin O, which is the only fixed point (and attractor).

**Figure 3 biology-09-00134-f003:**
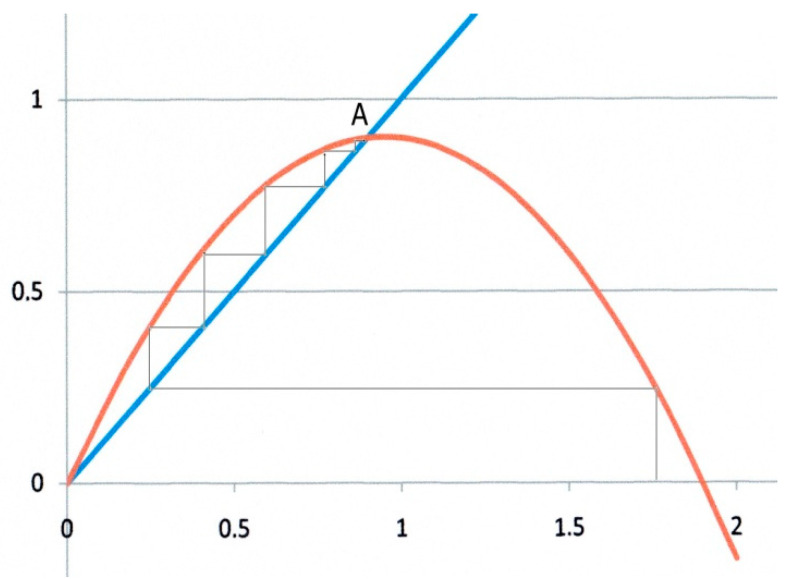
Logistic map for *α* = 0.9, *β* = 1.O is now a repulsor, and the system converges monotonically towards the attractor at *x** = 0.9.

**Figure 4 biology-09-00134-f004:**
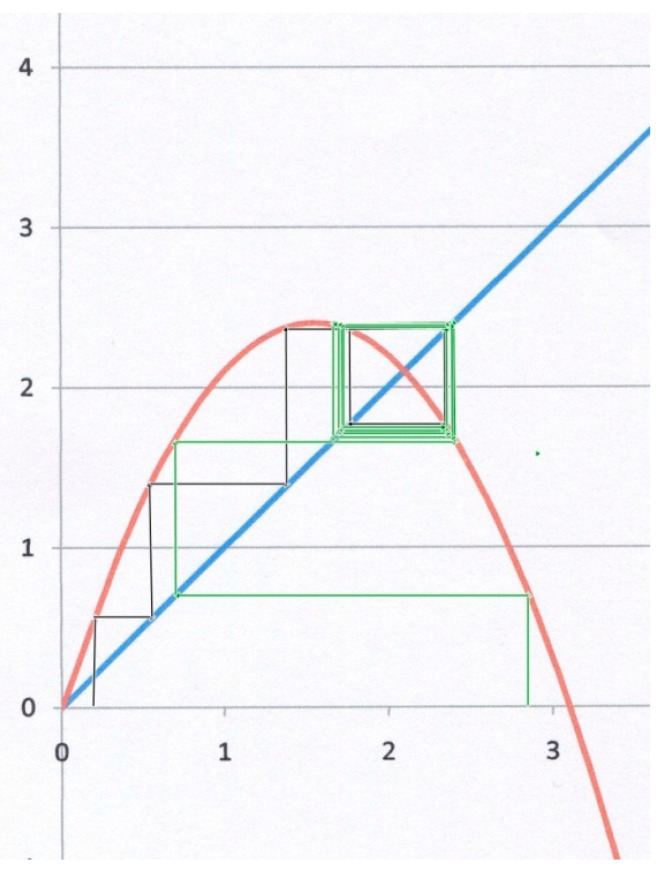
Logistic map for *α* = 2.1, *β* = 1. The slope of the parabola at *x* = *x** (which is still an attractor) is now less than −1, leading to a bi-stable limit cycle, whatever the starting point (black or green staircase lines).

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
