# Peer review of "A Brief Theory of Epidemic Kinetics"

_biology, 2020, doi:10.3390/biology9060134_

Round 1

Reviewer 1 Report

The author employed the theory of dynamical systems to quantitatively predict the temporal evolution of the COVID pandemic. The manuscript is worth of rapid publication. However, there are several points to clarify.

Line 15: What is the C/D ratio?

Line 28: What is the XX?

Line 50: What are the reasons for confining the study to the conditions to bring the reproduction factor below 1?

Line 69: What does the coefficient theta mean?

Line 108: Typo, htiscase -> this case

Figure 1: According to the manuscript, dNc/dt =0, when Nc=N. However, that is not consistent with Eq (2). When Nc=N, dNc/dt would be CN^2(1-θ) according to the equation.

Finally, it is desirable that the author creates Materials and Methods section, where the author should describe how the graphs were produced with sufficient detail including parameter values to allow others to replicate. In addition, please provide the name of software used for the calculations and production of the graphs.  

Reviewer 2 Report

Based on the theory of dynamical systems, the authors tried to summarize the brief theory of epidemic kinetics on COVID-19 pandemic.

I have some suggestion for them,

  1. The title must change to show the epidemic kinetics related to COVID-19, such as “A brief theory of epidemic kinetics related to COVID-19” or others that authors think better.
  2. Introduction, line 2, the authors mentioned about "Spanish Flu" in 1918-1919. I think it is better to mention about SARS that was more related to COVID-19.
  3. Furthermore, the epidemic growth kinetics theory or model may be proved or validate tested by using the data of SARS.
  4. The theory need to validate, and I suggest to use the previous endemic or pandemic data, such as "Spanish Flu", SARS, Avian flu, …
  5. The authors could use the data that happened in China, South Korea, Taiwan to test the theory, and may predict it in the USA, or Italy, or Germany…

Reviewer 3 Report

This paper presents a simple model for studying the expansion of covid19. The author claims the model incorporates the following characteristics: contagious and exposed people, contact duration, protection measures, and immunological response. In my opinion, the model was obtained by fundamental analysis of nonlinear dynamics using the well know theory of the SEIR model (Susceptible, Exposed, Infected, Recovered) and logistic models. Therefore, not new insights can be obtained from the proposed model since it follows the same spreading patterns. Also, a strong quantitative analysis is needed to support the hypotheses. Besides, the motivation of the paper is not clear; what new knowledge can be derived from the model? A comparison with published recent models or with real data is a must. Finally, the quality of the figures needs to be improved as well as the list of references must be enhanced by adding new and updated citations of the state of the art.

Round 2

Reviewer 2 Report

I have no more comments, since the previous comments were not accepted by the author. In addition, the author explained the changes of the revised manuscript, which I can accept.

Author Response

I thank Reviewer 2 for accepting the changes I made, along with my explanations.

Reviewer 3 Report

The author has justified the contribution of his paper. Thanks. On the other hand, please make sure the quality of the figures will be improved by the Editorial as you mentioned.

Author Response

Thank you for your comments.

About the quality of figures, they are not intended to show high resolution experimental data. Instead, they display simple smooth curves (straight lines and parabolas) directly obtained from my analytical equations. A higher resolution would not bring any additional information. Excel curve display standards seemed to me good enough for this purpose.

Nevertheless, I transferred the obtained Excel figures into "Photoscape" which helps introduce clear text boxes, and converted them into jpeg format, which would help the editorial staff to further improve them if necessary.